# Partial Biodegradable Blend with High Stability against Biodegradation for Fused Deposition Modeling

**DOI:** 10.3390/polym14081541

**Published:** 2022-04-11

**Authors:** Muhammad Harris, Hammad Mohsin, Johan Potgieter, Kashif Ishfaq, Richard Archer, Qun Chen, Karnika De Silva, Marie-Joo Le Guen, Russell Wilson, Khalid Mahmood Arif

**Affiliations:** 1Massey Agrifood Digital Lab, Massey University, Palmerston North 4410, New Zealand; j.potgieter@massey.ac.nz (J.P.); r.wilson@massey.ac.nz (R.W.); 2Industrial and Manufacturing Engineering Department, Rachna College of Engineering and Technology, Gujranwala 52250, Pakistan; 3Department of Polymer Engineering, National Textile University, Faisalabad 37610, Pakistan; mhammad@ntu.edu.pk; 4Industrial and Manufacturing Engineering Department, University of Engineering and Technology, Lahore 54890, Pakistan; kashif.ishfaq@uet.edu.pk; 5School of Food and Advanced Technology, Massey University, Palmerston North 4410, New Zealand; r.h.archer@massey.ac.nz (R.A.); q.chen2@massey.ac.nz (Q.C.); 6Faculty of Engineering, University of Auckland, Auckland 1023, New Zealand; k.desilva@auckland.ac.nz; 7Scion, Rotorua 3046, New Zealand; mariejoo.leguen@scionresearch.com; 8Department of Mechanical and Electrical Engineering, SF&AT, Massey University, Auckland 0632, New Zealand; k.arif@massey.ac.nz

**Keywords:** fused deposition modeling, additive manufacturing, polypropylene, polylactic acid, biodegradation, pellet, 3D printing

## Abstract

This research presents a partial biodegradable polymeric blend aimed for large-scale fused deposition modeling (FDM). The literature reports partial biodegradable blends with high contents of fossil fuel-based polymers (>20%) that make them unfriendly to the ecosystem. Furthermore, the reported polymer systems neither present good mechanical strength nor have been investigated in vulnerable environments that results in biodegradation. This research, as a continuity of previous work, presents the stability against biodegradability of a partial biodegradable blend prepared with polylactic acid (PLA) and polypropylene (PP). The blend is designed with intended excess physical interlocking and sufficient chemical grafting, which has only been investigated for thermal and hydrolytic degradation before by the same authors. The research presents, for the first time, ANOVA analysis for the statistical evaluation of endurance against biodegradability. The statistical results are complemented with thermochemical and visual analysis. Fourier transform infrared spectroscopy (FTIR) determines the signs of intermolecular interactions that are further confirmed by differential scanning calorimetry (DSC). The thermochemical interactions observed in FTIR and DSC are validated with thermogravimetric analysis (TGA). Scanning electron microscopy (SEM) is also used as a visual technique to affirm the physical interlocking. It is concluded that the blend exhibits high stability against soil biodegradation in terms of high mechanical strength and high mass retention percentage.

## 1. Introduction

The invention of additive manufacturing (AM) or 3D printing dates back to the 1990s and has introduced various new domains of research [1]. These include new AM technologies [2,3], materials [4,5], processing (pre, in situ, and post) [6], and applications [7]. Each new AM technology is designed for a particular material. For example, the fused filament fabrication (FFF) or fused deposition modeling (FDM) introduces the 3D printing of thermoplastics and elastomeric thermoplastics [8]. Stereolithography (SLA) registers its invention for thermoset polymers [9]. Selective laser sintering (SLS) emerges as another AM technique for thermoplastics but in fine powder form [10]. Direct metal laser sintering (DMLS) enlists the 3D printing of metals [11,12].

Among all abovementioned AM techniques, FDM is ranked high in terms of economic viability, processing, diversity in materials, and availability [13]. Furthermore, FDM is the only AM technique that encompasses the additive manufacturing of real-life products ranging from a few millimeters [14] to meters [1], making it the only AM technique to register large-scale thermoplastic products [1]. The materials used for large-scale products includes acrylonitrile butadiene styrene (ABS), which is reinforced with synthetic fibers (glass or carbon) [15]. Despite the commendable efforts in terms of research novelty, the overall mechanical endurance is not considerable for large-scale applications. Moreover, the rising concerns associated with harmful effects of nonbiodegradable polymers on the environmental ecosystem [16] is also a valid reservation for the use of reported materials (ABS).

Concerning one part of the problem, there are various biodegradable polymers that can be potential contenders for consideration, for example, polylactic acid (PLA) [17], polycaprolactone (PCL) [18], polyhydroxy alkanoate (PHA) [19], polyhydroxy valarate (PHV) [20], polybutylene succinate (PBS) [21], and chitosan (CS) [18]. Unfortunately, most of the abovementioned biodegradable polymers report a low mechanical stability of below 40 MPa (tensile strength) [22,23], apart from PLA that ranges above 80 MPA [24]. Therefore, PLA can be a potential candidate for large-scale additive manufacturing applications. However, the intermolecular shortcomings of PLA cause a drastic degradation in mechanical properties in real life environments such as soil [25], moisture [26], and thermal aging [27]. In this regard, PLA has been noted to undergo vigorous chain breakdown in real environments [25].

One of the simplest ways to raise the mechanical endurance is to melt-blend PLA with high-temperature nonbiodegradable polymers [28]. However, the main hindrance is the use of nonbiodegradable polymers in a large percentage of the total constituents of the blend. In this regard, all proposed blends of PLA report at least 20% of high-temperature nonbiodegradable polymers for achieving optimal mechanical properties [28,29]. Furthermore, these blend polymers have not yet been investigated for endurance against enzymatic or soil degradation (biodegradation).

The authors of this research have presented an overall solution to the aforementioned issues in the form of a novel polymer blend of polylactic acid and polypropylene with the lowest possible composition of nondegradable polymer [30]. The blend is partially compatibilized with the addition of a compatibilizer, i.e., polyethylene graft maleic anhydride (PE-g-MAH) [30]. The material is designed with intended high physical interlocking to achieve high resistance to moisture and thermal aging [30]. However, the novel materials have not yet been analyzed for biodegradation in a real environment (soil), which can be a potential weak aspect considering the large-scale applications.

Therefore, this research presents a polymer blend of a biodegradable PLA, designed with excessive physical interlocking, to be analyzed for thermo-mechanical and biodegradation. The research includes a detailed statistical design of experiment (DoE) for analyzing the statistical significance of various variables such as thermal, mechanical, and soil degradation. The results are discussed using multiple chemical (FTIR), thermochemical (DSC, TGA), and visual (SEM) techniques.

## 2. Materials and Methods

### 2.1. Materials

The IngeoTM PLA 2002D from NatureWorks was provided by the material science department, SCION, New Zealand. The specific weight of the 2002D was 1.24 g/cm^3^. The compatibilizer, PE-g-MAH (95:5), was procured from Shenzhen Jindaquan Technology, China. Moplen HP400N polypropylene with a specific weight of 0.91 g/cm^3^ and flow index of 11 g/10 min was provided by TCL-Hunt, New Zealand. The main aim of this research was to achieve the maximum possible physical interlocking alongside partial chemical grafting; therefore, a high MFI grade of PP was particularly selected for blending [29,31].

### 2.2. Melt Blending

An HST China-based thermostat blast oven was used to dry PLA, PP, and PE-g-MAH. The drying was performed for approximately 1 h. A HAAKE^TM^ rheomex single screw extruder (SCION, Rotorua, New Zealand) was used for the melt blending of all polymers. The melt blending resulted in a filament of 1.1 ± 0.2 mm in diameter that was pelletized into cylindrical pellets of 1.0 ± 0.2 mm length. Twin-screw extrusion has been reported with unwanted material degradation due to thermal shearing [32], which cannot be afforded in this research. Furthermore, the blend would be 3D-printed in a screw-based pellet 3D printer that may affect the original blend properties. Therefore, a single screw extruder was utilized for preparation of the blend.

The composition of the blend was based on the following three factors: (1) minimum nonbiodegradable polymer, (2) maximum physical interlocking, and (3) successful 3D printing of prepared composition.

Therefore, the blend compositions were continuously prepared until successful 3D printing was achieved. In this regard, the minimum reported composition of nonbiodegradable polymer (PP) was 20% [33,34,35,36,37] with a maximum of 75% of biodegradable constituent (PLA) [35,36,37]. The first composition (75:5:20) resulted in unwanted extrusion swelling (die swelling), as shown in Figure 1. The swelling was probably caused by a high percentage of compatibilizer (5%) due to excessive maleic anhydride [38]. The first composition was thus rejected due to the unsuitable extruded diameter (2.3 mm) of extrudate for 3D printing. Therefore, the second blend composition was decided with the lowest nonbiodegradable polymer (7.5%) and compatibilizer (0.5%) based on no signs of swelling in the literature [37]. The second composition (92:0.5:7.5) was successfully extruded with the desirable extrudate diameter. The compositions are provided in Table 1.

Based on the above-mentioned decided criteria for the preparation of compositions, the third composition was not prepared, as the second composition was successfully extruded and 3D-printed.

### 2.3. Pellet 3D Printing

Three-dimensional 3D printing or fused deposition modeling was performed on an in-house-built pellet 3D printer [39], as shown in Figure 2. The pellet 3D printer is built with a single lead screw that feeds the material into the heated barrel and also extrudes out melted material out of the 0.2 mm nozzle [39]. Unlike conventional lead screws in extruders, the lead screw in the customized pellet printer is designed with only one physical configuration, i.e., feeding [39]. The lead screw does not have any metering and compression configuration as found in normal injection molding or filament extruders [39]. The single physical feeding phase has been reported with low to negligible thermochemical shearing during 3D printing [32,39,40].

For this particular research, a few modifications were made, such as a selective laser sintering-based cone, Teflon insulative plate, and efficient area of slotted fluid channels (Figure 2). The three abovementioned modifications aimed to avoid pre-heating of the novel polymer blend before reaching the heating barrel. Therefore, these modifications enabled the printer’s capability to avoid pre-thermal degradation of the novel blend during 3D printing.

The computer-aided design (CAD) drawings were prepared in “SolidWorks version 2019” and saved in standard tessellation language (stl) format. The stl files were encoded into G-codes using slicing software (Slic3r) followed by 3D printing on the pellet 3D printer using “Pronterface”.

Each new material required its own set of processing variables (parameters). These 3D processing parameters were set in the slicing software. In this regard, numerous combinations of different variables were investigated while making the layer thickness (0.2 mm) [41], extrusion width (0.3 mm) [41], nozzle diameter (0.4 mm), infill pattern (45/−45) [41], and infill density (100%) [41] as constants. The set of optimal values for the investigated variables are given Table 2.

### 2.4. Soil Biodegradation Testing

Soil burial was performed to analyze the biodegradation (Figure 3). The ASTM standard (D638 type IV) [42] of neat PLA and the blend was printed as per the literature [30]. Three samples were prepared for characterization at most. The weight of each as-prepared sample was measured and denoted as “$m_{0}$”. The samples were buried in real soil at a depth of ≈1 m [43,44] for soil degradations analysis in Palmerston North (Figure 3). The geographical coordinates for burial location of the samples were longitude 40°22′47.0″ and latitude 175°36′49.9″. The samples were taken out after 45 days, washed with water, dried with paper towels, and left for 2 weeks for acclimatization at 25 ± 3 °C. The washing, drying, and acclimatization helped to remove soil debris/particles, additional moisture, and the settlement of sample temperature to room temperature, respectively. The weight of acclimatized dog bones was measured and denoted as “$m_{1}$”. The weight retention factor ($m_{R}$) was calculated using the following relation [43],
(1)$$m_{R}=\frac{m_{1}}{m_{0}}\times 100\%$$

The effects of soil degradation were also analyzed in terms of mechanical strength. A “general full factorial ANOVA with multiple levels” was used to analyze the effects of soil degradation on different levels of adhesion. As with soil burial treatment, different levels of adhesion were achieved with the help of variable bed and printing temperatures. Each combination of bed and printing temperature in the ANOVA design of experiment provided a different resistance to soil degradation. In this regard, three levels for both variables (bed and printing temperature) were selected based on ranges mentioned in the previous section (pellet 3D printing). The factors (parameters) and corresponding levels are given in Table 3. The combinations were analyzed for three samples at most.

The ANOVA DoE analyzed the results of tensile strength using statistical means with the help of confidence level (alpha “α”). The 95% confidence level was selected, meaning only a 5% probability of the statistical model obtaining different mean tensile values of the 18 combinations, as shown in Table 4. Different tensile strength values are interpreted as a difference in the effects of the combination of bed temperature, printing temperature, and soil treatment.

### 2.5. Mechanical Testing

The mechanical testing (tensile) was performed on an Instron 5967 with a load cell capacity of 30 kN. The characterization was performed using a contact type clip-on-gauge extensometer with a 25 mm span length. The rate of extension for characterization was set at 5 mm/min.

### 2.6. Fourier Transform Infrared Spectroscopy (FTIR)

Intermolecular Chemical interactions were analyzed using a Thermo electron Nicolet 8700 FTIR spectrometer. OMNIC E.S.P 7.1 was used to perform the postprocessing of spectrums. The postprocessing included normalizing and baseline correction. The analysis aimed to detect the probable chemical interactions between different functional groups or elements as a result of polymer blending, 3D printing in-process thermal variables, and soil biodegradation. The analysis was performed in transmittance mode using an average of 30 spectrums measured in the wavelength range of 400–4000 cm^−1^.

### 2.7. Differential Scanning Calorimetry (DSC)

Thermochemical analysis was performed on a NETZSCH 449-F1 Jupiter simultaneous thermal analyzer. The analysis aimed to obtain information associated with: (1) nature of chemical blending (physical interlocking or not), and (2) effects of soil biodegradation. The machine was operated in the temperature range of 25 °C–550 °C with a nitrogen purging of 50 mL/min. The rate of temperature increase was set at 10 °C/min.

### 2.8. Thermogravimetric Analysis (TGA)

Thermogravimetric analysis was also conducted in a NETZSCH 449-F1 Jupiter simultaneous thermal analyzer. The aim of the analysis was to obtain further information regarding physical interlocking and stability against soil biodegradation. The analyzer was operated within the range of 25 °C–550 °C under nitrogenous atmospheric conditions purged at 50 mL/min. The temperature was achieved in the main chamber at a rate of 10 °C/min.

The analysis of neat PLA was also performed for comparison with soil-degraded blends at two extreme printing combinations, i.e., “161 °C, 25 °C” and “171 °C, 85 °C”. The analysis was performed in terms of mass loss in percent of the total mass of the original sample (≈15 g). In this regard, the corresponding temperatures for different mass losses (50%, 60%, 70%, 80%, 90%, 92%, and 95%) were noted and compared with neat PLA.

### 2.9. Scanning Electron Microscope (SEM)

Visual analysis of polymer blending and layer adhesion was performed in a Hitachi TM3030 Plus desktop SEM. The backscattered electrons (BSE) mode was applied to analyze different samples. Another main aim of SEM was to validate the results obtained in FTIR, DSC, and TGA analysis.

## 3. Results

### 3.1. Soil-Based Biodegradation on Weight Retention

The effects of soil degradation on the weight retention of neat PLA and the blend are shown in Figure 4a.

For treated (soil-degraded) samples, the combinations with a bed temperature of 85 °C present the highest weight retention percentage. For neat PLA (soil-degraded) samples, some of the highest weight retentions are observed as 99.79% and 99.78% for combinations of (166, 85) and (171, 85), respectively. This shows mass degradations of 0.21% and 0.22%, after 45 days of soil treatment for treated neat PLA printed at combinations of (166, 85) and (171, 85), respectively.

For treated blend (soil-degraded) samples, the values of 99.89%, 99.88%, and 99.87% are observed for combinations of (161, 85), (166, 85), and (171, 85), respectively. This shows mass degradations of 0.11%, 0.12%, and 0.13% after 45 days of soil treatment for treated blend combinations of (161, 85), (166, 85), and (171, 85), respectively.

Overall, the weight retention ability improves with the increase in printing and bed temperature for neat PLA and the blend (Figure 4a). The neat PLA shows a significantly low weight retention (high biodegradation) as compared to the blend at all printed combinations. As a comparison, the range of 99.55–99.79% is observed for neat PLA for all combinations as compared to 99.89–99.76% of the treated blend. The high weight retention of the blend presents the high stability to soil degradation as compared to neat PLA.

### 3.2. Soil-Based Biodegradation on Tensile Strength

The ANOVA analysis shows the effects of soil degradation on the tensile strength. The general full factorial ANOVA-based design of experiment is designed with 18 combinations, as shown in Table 4.

The tensile strength against all designed combinations is reported in Table 4. For treated blend combinations, the highest tensile strength of 42.79 MPa is noted for combination “25 °C, 171 °C” followed by 39.5 MPa of combination 55 °C, 161 °C. For nontreated blend combinations, the highest strength of 44.95 MPa is reported for “85 °C, 161 °C”. This shows that the soil degradation does not result in a significant deterioration in tensile strength.

The statistical analysis reveals the printing temperature as the significant variable with a *p*-value of 0.032 (<“α = 0.05”) in Figure 4b. The printing temperature is further confirmed by ANOVA analysis in the “main-effects plot” (Figure 4c) with the printing temperature of 171 °C being reported as the optimal temperature for high strength. The *p*-values for the remaining two nonsignificant variables are 0.937, 0.063 for bed temperature and soil treatment, respectively (Figure 4d).

The soil degradation treatment is detected as insignificant (*p*-value = 0.063) with a small decrease of just 3 MPa (40 MPa to 37 MPa) in the “main-effects plot” (Figure 4c). The small decrease in tensile strength is supported by high weight retention, which highlights the stability of PLA/PP/PE-g-MAH against soil degradation. The Minitab analysis is provided in “Appendix A”. The details of the design of experiment (DoE) and the corresponding tensile strength are presented in Table 4.

## 4. Discussion

### 4.1. Analysis for Intermolecular Interactions

The FTIR spectrum of all neat polymers are confirmed with the literature [45,46,47]. Figure 5 shows the FTIR spectrum of neat PLA and PP.

The spectrums for the nontreated (nonbiodegraded) blend printed at the low-temperature combination (161 °C, 25 °C) and high-temperature combination (171 °C, 85 °C) are compared with the neat PLA (Figure 5). The spectrums reveal significant variations in the form of intermolecular interactions. For example, the shift in wave number is noted for C-H [48], C-O-C [48], and C=O functional groups [48] from 2997 cm^−1^ to 2989 cm^−1^, 1085 cm^−1^–1082.2 cm^−1^ to 1083 cm^−1^, and 1747 cm^−1^ to 1743 cm^−1^ respectively.

The intensities in percentage of numerous functional groups in the nontreated blend (161 °C, 25 °C) are also varied after melt blending as compared to neat PLA. For example, a 5% increase in C=O groups [48] is noted in nontreated blends (161 °C, 25 °C) as compared to neat PLA (90%). The significant rise in C=O is due to the observed intermolecular synchronization of similar groups of two different polymers. The synchronization is highlighted in Figure 5 with a magnified image of the hump that includes the C=O groups of both PLA and maleic anhydride at 1705 cm^−1^.

Another sign of intermolecular interactions is noted as a new C-H peak (2950 cm^−1^) in the nontreated blend (161 °C, 25 °C), which is not found in neat PLA (Figure 5). The new C-H peak is merged with three peaks of the nontreated blend (161 °C, 25 °C) originally inherited by the polypropylene. However, the interesting fact is the prominent reduction in intensity (98%) of new the C-H peak as compared to the one found in neat PLA (88%). The literature interprets the new distinct peak as a physical interlocking or phase separation and decrease in intensity as obstructed intermolecular movement [48].

The abovementioned FTIR confirms the melt blending to form physical interlocking along with signs of chemical interactions.

The comparison of soil-degraded samples at low- and high-temperature combinations is analyzed with the nontreated combination at the corresponding temperatures (low or high) in Figure 5. The low-temperature comparison reveals a decrease of about 2.3% (87.3–85%) of C-O-C and 5.5% (90.6–85%) of C=O groups in soil-degraded samples. The small depletion (2.3%) of C-O-C and high depletion of C=O show low chain scission and high chemical grafting [48], respectively, which provide stability to the novel blend against soil degradation. This is also verified with the insignificance of soil degradation in ANOVA analysis (Figure 4). On the contrary, the high printing temperatures (171 °C, 85 °C) decrease the C-O-C groups by 7.9% (95.9–88%) and the C=O by 5.5% (96.7–91.2%), as shown in Figure 5. The drastic decrease in C-O-C groups presents severe chain scission. The chain scission describes the decrease in tensile strength noted in the “main-effects plots” with the increase in printing temperature for soil-degraded samples (Figure 4). However, it is necessary to find more evidence to investigate the in-depth relation of chain scission with tensile strength.

Until this point of discussion, the physical interlocking and chemical grafting stand out as notable phenomena. However, the analysis requires DSC to analyze the exact nature of the polymer blend.

### 4.2. Analysis for Nature of Blending and Effects of Degradation Mechanisms

The analysis for effects of melt blending is performed on thermographs of neat PLA and as-prepared blend pellets (nonprinted), as shown in Figure 6. Two notable variations are observed, i.e., glass transition and melt crystallization.

Regarding the glass transition phase, the T_G_ of the as-prepared blend pellets is increased to 63.2 °C as compared to the 59.89 °C of neat PLA (Figure 6). The enthalpy of glass transition is also noted with a significant increase (≈1.7 J/g) for blend pellets as compared to neat PLA (0.026 j/g). The increases in T_G_ and ΔH_G_ present the re-orientation of polymeric chains and thus show the improved formation of crystallites [49].

Regarding the melt crystallization phase, the as-prepared pellets are found with a bimodal peak (Figure 6). The multiple peaks validate the phase separation or the physical interlocking [33] of PP in the PLA matrix. Furthermore, the T_M_ of the as-prepared blend pellets (155.5 °C) is observed with a minor increase as compared to 154.7 °C of polylactic acid, which is probably due to chemical grafting [48].

The soil degradation after printing at the high-temperature combination (171 °C, 85 °C) shows a visible decrease in almost all parameters as compared to the low-temperature combinations (161 °C, 25 °C). For example, the ΔH_G_, ΔH_C_, ΔH_M_, and ΔH_D_ decrease to 1.57 J/g, 10.56 J/g, 9.75 J/g, and 647.5 J/g from 2.086 J/g, 13.94 J/g, 11.96 J/g, and 787.3 J/g, respectively. Similarly, the T_C_, T_M_, and T_D_ decrease to 104.1 °C, 156 °C, and 365.8 °C from 106.9 °C, 156.2 °C, and 369.8 °C, respectively. The decrease in properties is clearly related to the chain scission [48], as found in FTIR analysis (Figure 5). Furthermore, the decrease in the aforementioned DCS parameters for soil-degraded samples at high temperature provides a thermo-chemical justification for the two results obtained in ANOVA analysis: (1) significant printing temperature (Figure 4b) and (2) decrease in strength after soil biodegradation (Figure 4c).

Based on the decrease in thermal properties in DSC and chain scission in FTIR, a suitable reason for mechanical stability after soil degradation is the physical interlocking. However, it requires TGA analysis to further validate the effects of interlocking.

### 4.3. Measurement of Interlocking and Chemical Grafting

Thermogravimetric analysis (TGA) is used to validate the FTIR and DSC results regarding physical interlocking. The analysis also aims to analyze the thermal stability to the degradation after soil biodegradation and hydrolytic degradation.

The thermogravimetric analysis of neat PLA, PP, and printed blends is shown in Figure 7. The physical interlocking is confirmed from the distinct step that occurs above 400 °C. However, the mass percentage of the step associated with PP occurs at 6.19% and 6.75% for soil-degraded samples of “161 °C, 25 °C” and “171 °C, 85 °C”, respectively. Both (6.19% and 6.75%) are less than the added percentage of PP in the blend (7.5%). The mass percentage less than 7.5% shows minor chemical grafting [48]. Hence, the desired characteristics of the blends are achieved and validated.

The soil-degraded samples at the low-temperature combination (161 °C, 25 °C) reveals a near-similar onset temperature (348.9 °C) as compared to 350.3 °C of neat PLA (Figure 7). The T_END_ for most degradation percentages (50% to 90%) is also noted similar with a maximum of 0.59% difference for 70% mass loss (Table 5). The high-temperature combination (171 °C, 85 °C) provides a near-3% difference in the temperature for mass losses of 50% to 70% (Table 5). The decrease in temperature in TGA is explained by (1) the chemical chain scission in FTIR (Figure 5), and (2) the decrease in thermal parameters (T_M_, T_D_, ΔH_G_, ΔH_C_, ΔH_M_, ΔH_D_) in DSC (Figure 6). Based on the aforementioned decrease in different parameters of the FTIR, DSC, and TGA results, the drop (but insignificant) in tensile strength with the soil degradation in the “main-effect plots” of ANOVA (Figure 4) is thermochemically verified. However, the statistically insignificant decrease proves the stability of the blend against soil degradation. A suitable reason for such a low strength loss is the physical interlocking of PP (6.19% and 6.75%) in the PLA matrix as found with a phase separation in TGA graphs (Figure 7).

### 4.4. Morphological Analysis Using SEM

Scanning electron microscopy (SEM) further confirms the physical interlocking through visual analysis (Figure 8). The blend appears with a clear phase separation of PP in the PLA matrix. The distinct PP fiber can be noted in Figure 8c. Therefore, the SEM analysis proves the overwhelming physical interlocking as mentioned in FTIR, DSC, and TGA analysis. The physically interlocked PP is the reason for high stability against biodegradation and moisture hydrolytic degradation.

## 5. Conclusions

This work presents the detailed analysis of effects of in-process temperatures and soil biodegradation on an FDM blend with excess physical interlocking and reasonable chemical grafting. The polymer blend stands out among contemporary FDM blend systems due to the lowest percentage (7.5%) of a nonbiodegradable polymer (PP) in a biodegradable polymer (PLA). The approach of physical interlocking helps to achieve high tensile strength and soil degradation. The research includes a detailed design of the experiment consisting of mixed-level general full factorial ANOVA. The DoE includes the variables capable of causing in-process thermal variations and post-printing biodegradation. In this regard, the bed temperature and printing temperature are selected as in-process thermal variables, and the interval of soil burial is considered as a variable for analyzing soil degradation. The statistical results are supported with post-mechanical characterizations including FTIR, DSC, TGA, and SEM. The research concludes with the following outcomes.

The novel blend system is statistically stable against 45 days of biodegradation with a *p*-value greater than 5% in confidence level.

The in-process 3D printing (nozzle) temperature is the significant variable with a *p*-value less than the 5% confidence level.

The blend reports the highest weight retention of 99.89% after 45 days of biodegradation, which is far higher than that of neat PLA. This shows high stability against biodegradation.

The FTIR reveals the chain scission of C-O-C bonds in the blend due to which the tensile strength shows a statistically insignificant decrease of 4 MPa, i.e., 42 MPa to 38 MPa.

The low decrease in tensile strength after biodegradation presents high stability against biodegradation. The reason for the observed stability, even with chain scission, is due to physical interlocking confirmed in DSC, TGA, and SEM characterizations. The physical interlocking of at least 6% is found for both low-temperature (161, 25) and high-temperature (171, 85) combinations.

## Figures and Tables

**Figure 1 polymers-14-01541-f001:**
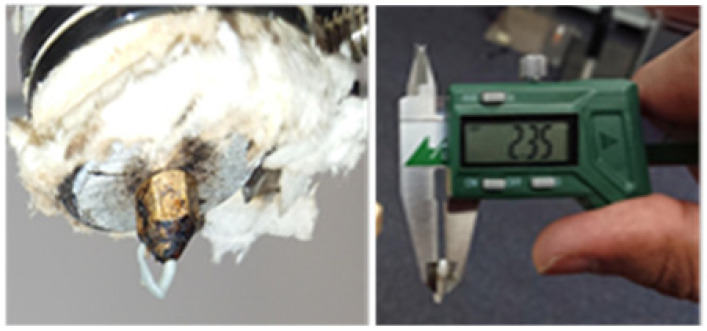
Die swelling in extruded filament from pellet printer.

**Figure 2 polymers-14-01541-f002:**
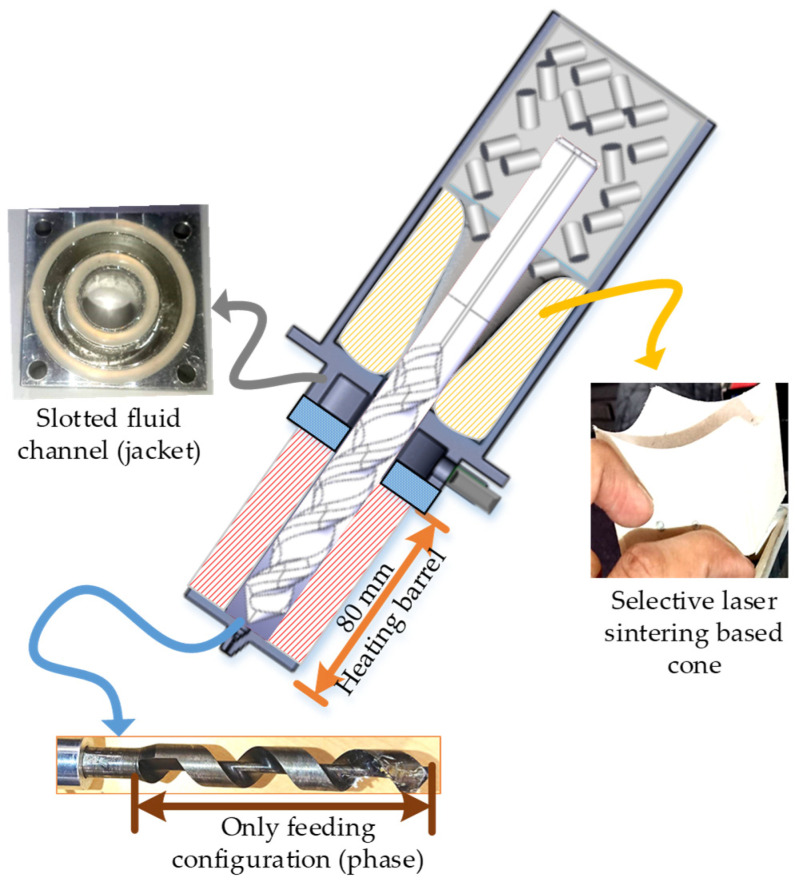
Pellet 3D printer with modifications [39].

**Figure 3 polymers-14-01541-f003:**
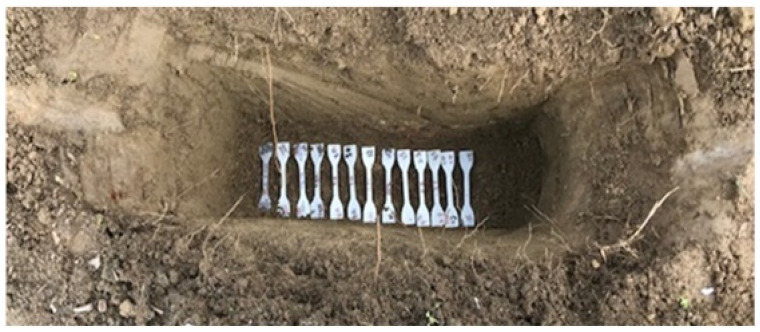
Location and orientation of buried samples for soil degradation.

**Figure 4 polymers-14-01541-f004:**
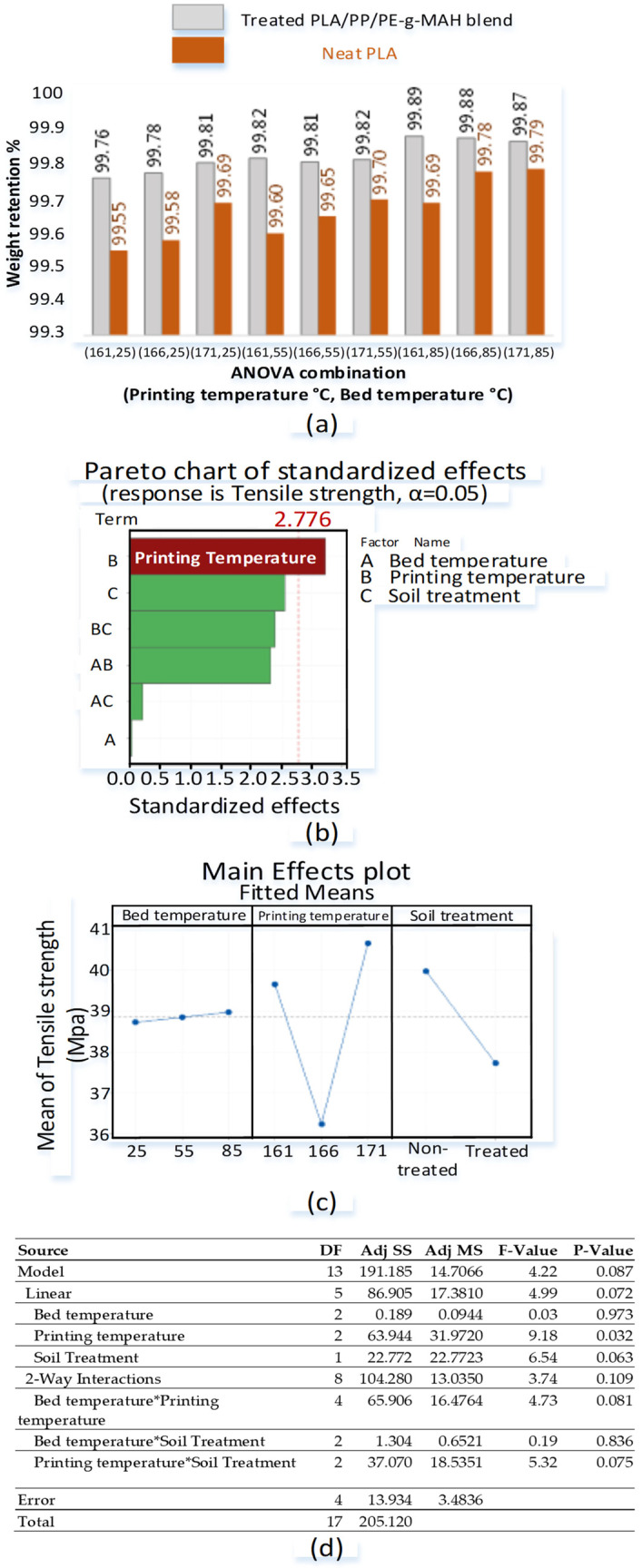
Results for soil degradation: (**a**) weight retention %, (**b**) pareto chart, (**c**) main-effects plots, and (**d**) ANOVA analysis.

**Figure 5 polymers-14-01541-f005:**
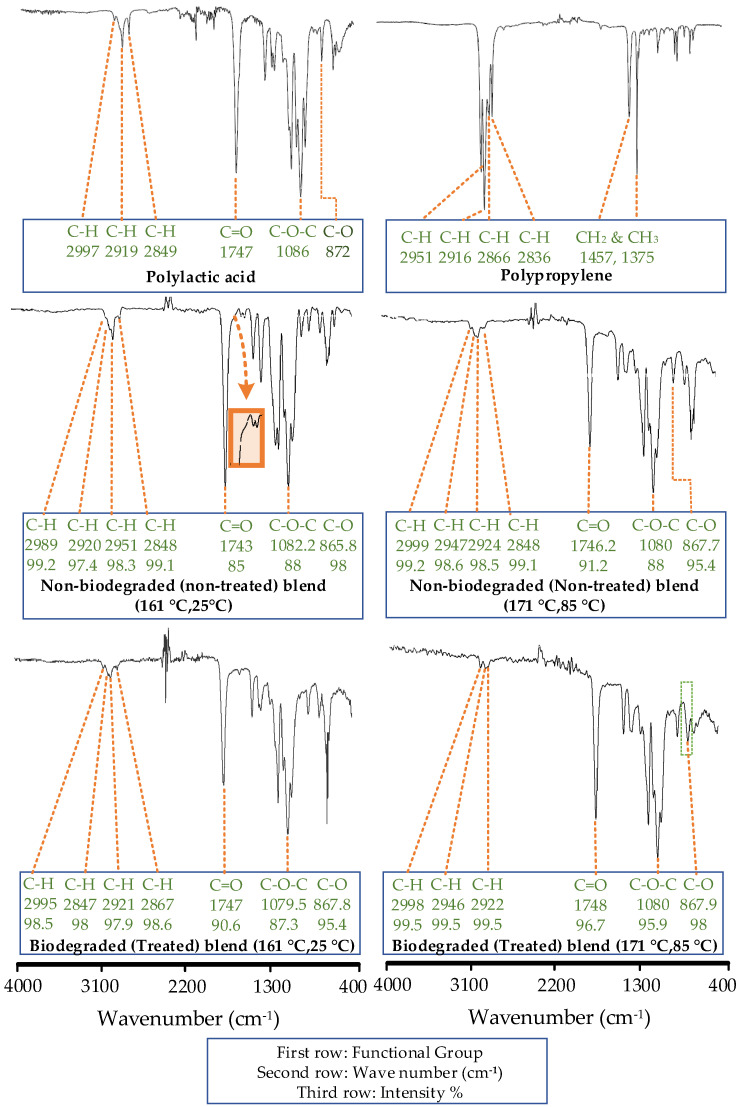
FTIR analysis of the effects of melt blending, 3D printing (non-treated), and 3D printing (treated).

**Figure 6 polymers-14-01541-f006:**
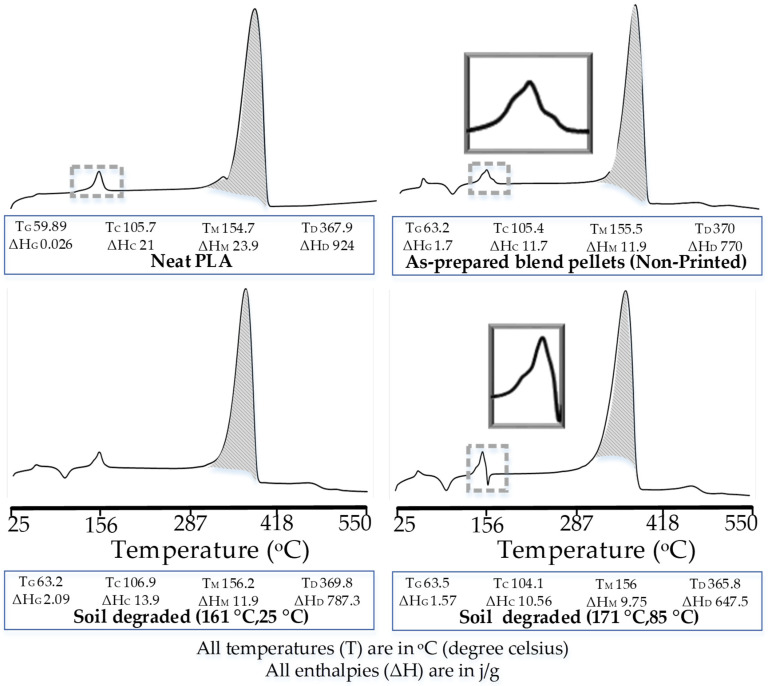
DSC analysis of neat PLA, as-prepared blend pellets, and soil-biodegraded samples.

**Figure 7 polymers-14-01541-f007:**
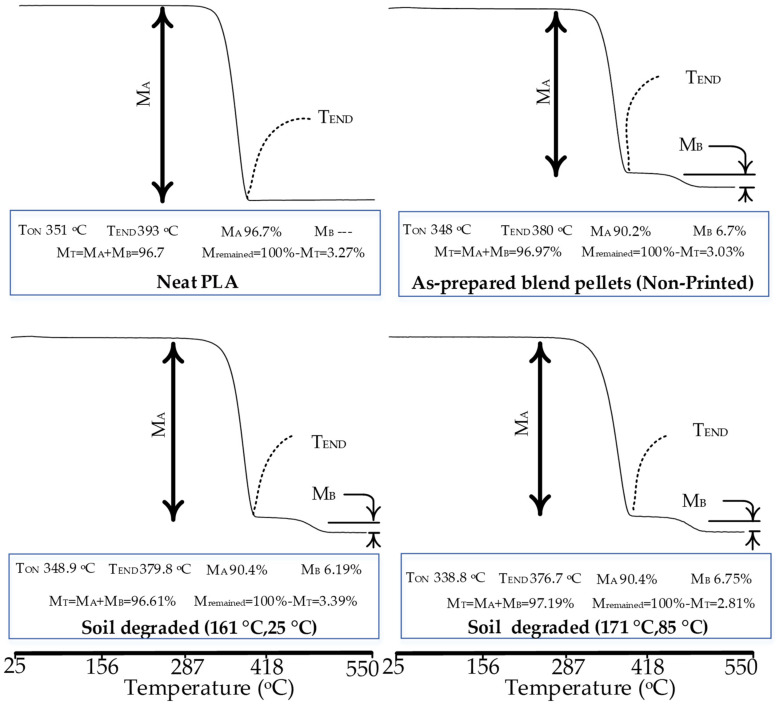
TGA analysis for physical interlocking and soil biodegradation.

**Figure 8 polymers-14-01541-f008:**
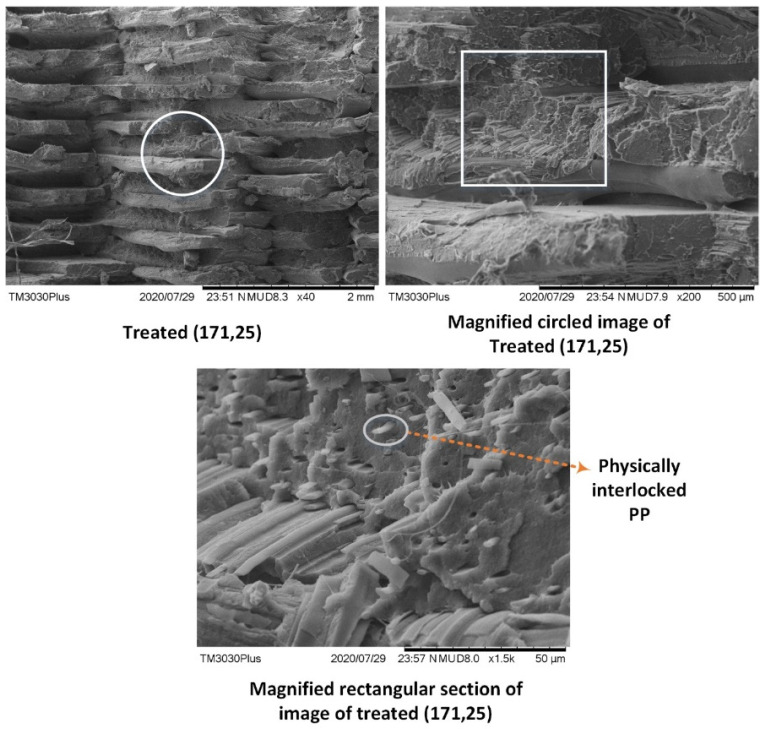
SEM analysis for PLA/PP/PE-g-MAH blend at 171 °C, 25 °C.

**Table 1 polymers-14-01541-t001:** Compositions prepared for ternary blend systems.

Blend	Biodegradable Polymer (PLA)	Compatibilizer	Nonbiodegradable Polymer (PP)	Extrudate Diameter (Must Be ≤0.2)	Decision
**1**	75	5	20	2.3 ± 0.05	Rejected and moved to next composition with less MAH and PP
**2**	92	0.5	7.5	0.2 ± 0.05	Successfully 3D-printed. Further compositions are not required.

**Table 2 polymers-14-01541-t002:** Optimal 3D printing variables.

Parameter	Set Value
Multiplier	5
Printing speed	15 m/min
Bed temperature	25 °C, 55 °C, 85 °C
Printing temperature	161 °C, 166 °C, 171 °C

**Table 3 polymers-14-01541-t003:** General full factorial design of experiment (DoE) for soil degradation analysis.

Factor (Parameter)	Level 1	Level 2	Level 3
Printing Bed (surface) temperature	25 ± 2 °C	55 ± 2 °C	85 ± 2 °C
Printing (nozzle) temperature	161 ± 3 °C	166 ± 3 °C	171 ± 3 °C
Soil burial interval	0 days	45 days	

**Table 4 polymers-14-01541-t004:** DoE for analysis of soil degradation effects on tensile strength.

StdOrder	RunOrder	PtType	Blocks	Bed Temperature	Printing Temperature	Soil Treatment	Tensile Strength (MPa)
4	1	1	1	25	166	Treated	37.15
6	2	1	1	25	171	Treated	42.79104
8	3	1	1	55	161	Treated	39.53389
5	4	1	1	25	171	Non-treated	43.37669
15	5	1	1	85	166	Non-treated	32.49289
1	6	1	1	25	161	Non-treated	38.92701
18	7	1	1	85	171	Treated	38.850825
13	8	1	1	85	161	Non-treated	44.959735
17	9	1	1	85	171	Non-treated	43.10712
11	10	1	1	55	171	Non-treated	40.01403
7	11	1	1	55	161	Non-treated	42.99
9	12	1	1	55	166	Non-treated	37.71559
16	13	1	1	85	166	Treated	36.9
2	14	1	1	25	161	Treated	33.936205
3	15	1	1	25	166	Non-treated	36.12446
12	16	1	1	55	171	Treated	35.73578
10	17	1	1	55	166	Treated	37.063755
14	18	1	1	85	161	Treated	37.5

**Table 5 polymers-14-01541-t005:** Temperatures for a particular mass loss. All percentages of biodegraded samples are calculated with respect to the corresponding temperature of neat PLA. The negative numbers designate the decrease in temperature and positive numbers designate the increase in temperature of biodegraded samples.

Mass Loss%	PLA	Soil 161 °C, 25 °C	Soil 171 °C, 85 °C
50%	368 °C	365.7	359.5
	0.0	−0.63	−2.31
60%	372 °C	369.1 (−0.57)	363.7 (−2.02)
	0.0	−0.78	−2.23
70%	375 °C	372.5 (−0.59)	367.8 (−1.84)
	0.0	−0.67	−1.92
80%	378 °C	376.3 (−0.50)	372.3 (−1.56)
	0.0	−0.45	−1.51
90%	383 °C	382.6 (+0.03)	379.3 (−0.84)
	0.0	−0.10	−0.97
92%	384 °C	419.6 (+9.36)	432.5 (+12.7)
	0.0	9.27	12.63
95%	386 °C	462.1 (+19.8)	463.3 (+20.1)
	0.0	19.72	20.03

## Data Availability

Not applicable.

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
