# Peer review of "Partial Biodegradable Blend with High Stability against Biodegradation for Fused Deposition Modeling"

_polymers, 2022, doi:10.3390/polym14081541_

Round 1
Reviewer 1 Report
The manuscript under review reports on research oriented on preparation and comprehensive analysis of partial biodegradable polymeric blend aimed for large-scale fused deposition modeling. In my opinion the work itself is well written, the topic studied is relevant and important due to the increasing use of 3d printing, thus it can be accepted for publications in the Polymers. However, some minor adjustments are needed before publication.
1) As far as I understood the work is a continuation of previous works (two?), anyhow the authors cite only position [30]. Please clarify it Parts 1&2 are already published, perhaps it would be better to change the title of the paper simply into: Partial biodegradable blend with high stability against 2
biodegradation for fused deposition modeling
2) The quality (resolution) of fig. 4 must be improved.
3) Subsection 3.1 and 3.2: it would be good to described in more details the results shown in fig. 4 and table 4; please add some more information regarding employed DoE methodology.
Author Response
Reviewers’ comments
We are grateful for valuable time by respected reviewers. We have incorporated the required data/modifications to all suggested comments. The details are given below,
The manuscript under review reports on research oriented on preparation and comprehensive analysis of partial biodegradable polymeric blend aimed for large-scale fused deposition modeling. In my opinion the work itself is well written, the topic studied is relevant and important due to the increasing use of 3d printing, thus it can be accepted for publications in the Polymers. However, some minor adjustments are needed before publication.
Reviewer comment:
1) As far as I understood the work is a continuation of previous works (two?), anyhow the authors cite only position [30]. Please clarify it Parts 1&2 are already published, perhaps it would be better to change the title of the paper simply into: Partial biodegradable blend with high stability against 2
biodegradation for fused deposition modeling
Authors modification/answer:
The title is modified as suggested.
Reviewer comment:
2) The quality (resolution) of fig. 4 must be improved.
Authors modification/answer:
The Figure 4 is now redesigned (Line 239) and added with additional part “d”.
Reviewer comment:
3) Subsection 3.1 and 3.2: it would be good to described in more details the results shown in fig. 4 and table 4; please add some more information regarding employed DoE methodology.
Authors modification/answer:
The subsections 3.1 and 3.2 are now added with detail as suggested. Furthermore, the Figure 4 is also added with “part d” and explained with detail as well.
The corresponding tensile testing results in section “3.1. Soil-based biodegradation on weight retention” are explained in lines 232 to 237 and 241 to 250.
The corresponding tensile testing results in section “3.2. Soil-based biodegradation on tensile strength” are explained in lines 256 to 266.
Additional Authors modification/answer:
Additionally, the self citations are now reduced to only required ones. Previously there were 6 self cited articles. Now there are only 3, which are needed to be incorporated into article.

Reviewer 2 Report
This work presents "Partial biodegradable blend with high stability against biodegradation for fused deposition modeling". A blend of polylactic acid (PLA) and polypropylene (PP) with different bed and printing temperatures was prepared and thermal, mechanical, and soil degradation properties of the samples were evaluated. The study is interesting, and the topic is current. The manuscript is recommended to be published after including and addressing the below-listed comments with major corrections.
-The soil biodegradation part must be elaborated. The method must be explained clearly. The definition of weight retention and mass loss (%) must be clearly explained. Also please explain Table 5. A very high temperature of 360 to 386 °C for PLA treatment was not mentioned in the soil biodegradation method.
- The quality of Figure 4 must be improved.
- Please write the sample size for tensile and soil degradation experiments.
- “treated samples” is missing in the caption of Figure 5.
Author Response
Reviewers’ comments
We are grateful for valuable time by respected reviewers. We have incorporated the required data/modifications to all suggested comments. The details are given below,
This work presents "Partial biodegradable blend with high stability against biodegradation for fused deposition modeling". A blend of polylactic acid (PLA) and polypropylene (PP) with different bed and printing temperatures was prepared and thermal, mechanical, and soil degradation properties of the samples were evaluated. The study is interesting, and the topic is current. The manuscript is recommended to be published after including and addressing the below-listed comments with major corrections.
Reviewer comment:
-The soil biodegradation part must be elaborated. The method must be explained clearly. The definition of weight retention and mass loss (%) must be clearly explained.
Authors modification/answer:
The method is now explained with more detail. The sample numbers, location of sample burial, and involved processes (washing, drying, acclimatization) are explained. Furthermore, the associated ANOVA combination in section “2.4. Soil biodegradation testing” is also explained in detail.
The corresponding tensile testing results in section “3.2. Soil-based biodegradation on tensile strength” are explained in lines 256 to 266.
Reviewer comment:
Also please explain Table 5. A very high temperature of 360 to 386 °C for PLA treatment was not mentioned in the soil biodegradation method.
Authors modification/answer:
The method is now mentioned in TGA methodology in lines 218 to 222. Kindly consider it in TGA section as it is being added in “Discussions”.
Reviewer comment:
- The quality of Figure 4 must be improved.
Authors modification/answer:
The Figure 4 is now redesigned (Line 239) and added with additional part “d”.
Reviewer comment:
- Please write the sample size for tensile and soil degradation experiments.
Authors modification/answer:
The samples numbers are now mentioned in line 165 and 183.
Reviewer comment:
- “treated samples” is missing in the caption of Figure 5
Authors modification/answer:
It is now mentioned in line 279.
Additional Authors modification/answer:
Additionally, the self citations are now reduced to only required ones. Previously there were 6 self cited articles. Now there are only 3, which are needed to be incorporated into article.
The title is modified as suggested by one of the respected reviewers.

Round 2
Reviewer 2 Report
Thanks for the corrections. The manuscript is ready to be published.